# Effects of Different Exercise Therapies on Balance Function and Functional Walking Ability in Multiple Sclerosis Disease Patients—A Network Meta-Analysis of Randomized Controlled Trials

**DOI:** 10.3390/ijerph19127175

**Published:** 2022-06-11

**Authors:** Zikang Hao, Xiaodan Zhang, Ping Chen

**Affiliations:** Department of Physical Education, Laoshan Campus, Ocean University of China, 238 Song Ling Rd., Qingdao 266100, China; haozikang@stu.ouc.edu.cn (Z.H.); zhangxiaodan@stu.ouc.edu.cn (X.Z.)

**Keywords:** rehabilitation, yoga, aquatic exercise, multiple sclerosis disease, network meta-analysis

## Abstract

The objective of this research is to assess the effects of seven different exercise therapies (aquatic exercise, aerobic exercise, yoga, Pilates, virtual reality exercise, whole-body vibration exercise, and resistance exercise) on the balance function and functional walking ability of multiple sclerosis disease patients. Materials and Methods: The effects of different exercise interventions on the balance function and functional walking ability in people with multiple sclerosis were assessed by searching five databases: PubMed, Embase, Cochrane Library, Web of Science, and CNKI; only randomized controlled trials were included. The included studies were assessed for risk of bias using the Cochrane assessment tool. Results: The RCTs were collected between the initial date of the electronic databases’ creation and May 2022. We included 31 RCTs with 904 patients. The results of the collected data analysis showed that yoga can significantly improve patients’ BBS scores (SUCRA = 79.7%) and that aquatic exercise can significantly decrease patients’ TUG scores (SUCRA = 78.8%). Conclusion: Based on the network meta-analysis, we suggest that although each type of exercise is useful, yoga, virtual reality training, and aerobic training are more effective in improving the balance function of people with MS; aquatic exercise, virtual reality training, and aerobic training are more effective in improving the functional walking ability of people with MS.

## 1. Introduction

Multiple sclerosis is one of the most common disabling neurological diseases worldwide and has an average age of onset of 29 years. As of 2017, there were 2.5 million people with multiple sclerosis worldwide, and this number is increasing [1,2]. The disease has many adverse effects on patients, including, but not limited to, physical symptoms such as muscle weakness and reduced mobility and balance, as well as mental symptoms such as fatigue and cognitive decline [3,4]. This has a considerable impact not only on the patients themselves and their families, but also on public health and safety [5].

Due to the combination of reduced physical and mental function, approximately 75% of people with MS experience balance and walking-related impairments in the early and later stages of the disease [6], which increases their risk of falls and injuries [1]. The physical injuries and psychological fears associated with falls may further affect patients’ physical and mental health, creating a vicious cycle that further affects patients’ quality of life [7,8].

More and more research is focusing on the rehabilitation of people with MS, and in addition to traditional measures such as daily care and rehabilitation, different forms of exercise are increasingly being used in clinical non-pharmacological treatment and rehabilitation [9]. A number of randomized controlled trial studies show that exercise produces beneficial effects on mental aspects such as fatigue and cognitive performance in patients that exceed those of traditional rehabilitation measures. In addition, there are meta-analyses comparing one exercise intervention versus traditional rehabilitation measures on the physical and mental functional abilities of MS patients [10,11], as well as network meta-analyses comparing multiple exercise interventions versus traditional rehabilitation measures on the mental function of MS patients [12,13]; both provide considerable clinical evidence-based recommendations.

In addition to traditional meta-analysis, researchers invented a new evidence-based medical technique, network meta-analysis (NMA), which, in contrast to the original technique, allows one to compare and rank the effects of multiple interventions for a disease at the same time [14]. Therefore, in this study, we use network meta-analysis to compare different exercise programs (aquatic exercise, aerobic exercise, yoga, Pilates, virtual reality exercise, whole-body vibration exercise, and resistance exercise) in order to assess the effect of these programs on the physical function of people with MS, and to provide patients and clinicians with appropriate evidence-based recommendations.

## 2. Materials and Methods

### 2.1. Search Strategy

The authors in this paper searched five electronic databases (Pubmed, EMBASE, the Cochrane Central Register of Controlled Trials, Web of Science, and CNKI) from their inception to May 2022. The search strategy was constructed using the PICOS tool: (P) Population; people with multiple sclerosis, (I) Intervention; exercise, (C) Comparator; control group with usual care and usual rehabilitation measures only, (O) Outcomes; motor function tests of people with multiple sclerosis, and (S) Study type; RCTs [15]. The detailed search strategy is shown in Table 1 (using Pubmed as an example).

### 2.2. Inclusion Criteria

The inclusion criteria includes: (1) experimental group; use of an exercise as an intervention to treat multiple sclerosis disease, (2) control group; treatment of multiple sclerosis disease using only daily care and conventional rehabilitation (no other types of exercise interventions, just the more popular and commonly used balance rehabilitation exercises; no training, just daily living care), (3) clinical randomized controlled trial, and (4) outcome indicators including at least one of either the Berg Balance Scale (BBS) score or the Timed-Up-and-Go (TUG) score.

### 2.3. Exclusion Criteria

The exclusion criteria includes: (1) studies with incomplete or unreported data, and (2) studies from non-randomized controlled trials (including quasi-randomized controlled trials, animal studies, protocols, meeting abstracts, case report correspondence).

### 2.4. Outcomes

The primary outcome is the Berg Balance Scale (BBS), and the secondary outcome is the Timed-Up-and-Go (TUG) score. The BBS and TUG are popular with clinical practitioners due to their comprehensiveness, sensitivity, and simplicity [16,17].

### 2.5. Study Selection

Two authors independently screened the papers using Zotero software to eliminate duplicate papers and complete the primary screening. The titles and abstracts of the remaining papers were then read to eliminate those that did not meet the inclusion criteria, completing the re-screening process. Finally, the remaining papers were read in full to complete the final screening process, and the results were compared. Articles were included when two authors agreed that the inclusion criteria were met; if there was a disagreement between the authors regarding the inclusion of a paper, a third author was consulted to determine whether or not it should be included in the analysis.

### 2.6. Data Extraction

A table with seven sections [18] was used to extract detailed data from the included papers: (1) author (abbreviations), (2) year of publication, (3) country in which each study was conducted, (4) sample size in each study, (5) details of the experimental group, (6) details of the control group, and (7) outcome indicators.

### 2.7. Risk of Bias of Individual Studies

Two authors independently assessed the risk of bias (ROB) in accordance with the Cochrane Handbook version 5.1.0 tool (Cochrane, London, UK) for assessing ROB in RCTs. The following seven domains were considered: (1) randomized sequence generation, (2) treatment allocation concealment, the blinding of (3) participants and (4) personnel, (5) incomplete outcome data, (6) selective reporting, and (7) other sources of bias. Trials were categorized into three levels of ROB according to the number of components for which high ROB potentially existed: high risk (five or more), moderate risk (three or four), and low risk (two or less). All of the studies are, by default, classified as having a high ROB with respect to the category “blinding of participants” because it is impossible to blind participants to group assignments in exercise intervention protocols [19,20].

### 2.8. Data Analysis

First, because we are studying the efficacy of exercise for a particular disease, we chose to use continuous variables for statistical analysis. To calculate the results more conservatively, we used the immediate post-intervention value minus the baseline value to express the size of the intervention effect. As the results we analyzed were all in uniform units, we chose to use standard difference (SD) rather than standardized mean difference (SMD) for our calculations. There is bound to be variation between the original studies, and to make the results more scientific, we chose to calculate a random effects model rather than a fixed effects model [14].

Secondly, Stata software was used to present the network graphs, which are important in NMA. In a network diagram, different graphs have different meanings: (1) each node represents an exercise intervention; (2) the size of the node indicates the sample size of the subjects who performed this intervention; (3) if there are no line segments between each node, it means that indirect comparisons will be made between the nodes, and if there are line segments, it means that direct comparisons will be made between the nodes; (4) the thickness of the line segments between the nodes indicates the original study sample size; and (5) the size of the nodes and the thickness of the line segments are positively correlated with the number [21].

Again, we used Stata software to summarize and analyze the NMA using Markov chain Monte Carlo simulation chains in a Bayesian-based framework. Thus, in a ranking table, treatments were ranked from best to worst along the leading diagonal. Above the leading diagonal are estimates from pairwise meta-analyses, and below the leading diagonal are estimates from network meta-analyses [22,23].

Finally, we calculated the SUCRA ranking in Stata software and used it as a criterion for evaluating the effect of the exercise interventions, which is a percentage with a maximum value of 1 and a minimum value of 0. The closer to 1, the better the intervention effect; the closer to 0, the worse the intervention effect. A funnel plot will also be generated to examine possible publication bias [21].

## 3. Results

### 3.1. Study Identification and Selection

A total of 3744 articles were retrieved. A total of 525 articles were duplicates, and after eliminating them, 3219 articles remained. After screening the abstracts and titles of these 3210 articles, it was determined that 3041 articles were not relevant to the study. After reading the full texts of the 178 articles that remained after screening, 31 articles were finally included in the study. Refer to Figure 1 for specific details.

### 3.2. Characteristics of the Included Studies

We included 31 RCTs, with a total of 904 subjects in all of the studies combined. Interventions in the control group included Pilates training (seven studies) [24,25,26,27,28,29,30], whole-body vibration training (three studies) [31,32,33], aquatic training (two studies) [34,35], yoga training (two studies) [36,37], aerobic training (eight studies) [38,39,40,41,42,43,44,45], resistance training (three studies) [46,47,48], and virtual reality training (five studies) [43,49,50,51,52,53]. Sixteen studies were from Asia, four studies were from the Americas, and eleven studies were from Europe. The details of included studies are shown in Table 2.

### 3.3. Quality Assessment of the Included Studies

Only 13% of the studies had a high risk of bias. A total of 13 studies had a moderate risk of bias, and 12 studies had a low risk of bias. The overall risk of bias was acceptable, but it is worth noting that exercise as an intervention has the inherent disadvantage of it being difficult to implement a double-blind approach in an experiment. The specific risk of bias assessment scores for each study are presented in Appendix A.

### 3.4. Network Meta-Analysis

#### 3.4.1. Berg Balance Scale (BBS)

The results of the network meta-analysis showed that relative to the control group’s conventional rehabilitation measures, yoga (MD = 5.50, 95% CI = (2.55, 8.45)); virtual reality technology exercise (MD = 4.12, 95% CI = (2.15, 6.09)); aerobic exercise (MD = 4.01, 95% CI = (1.81, 6.20)); aquatic exercise (MD = 3.43, 95% CI = (0.23, 6.63)), and Pilates (MD = 2.70, 95% CI = (0.71, 4.69)) were superior compared to the control group in terms of increasing BBS scores (Table 3). Yoga achieved the number one SUCRA probability ranking in terms of increasing BBS scores (SUCRA: 79.7%). Details are shown in Figure 2. The *p*-values for the heterogeneity tests can be found in Appendix A.

#### 3.4.2. Timed-Up-and-Go Score (TUG)

The results of the network meta-analysis showed that relative to the control group’s conventional measures, aquatic exercise (MD = −2.58, 95% CI = (−5.88, −0.72)); aerobic exercise (MD = −1.53, 95% CI = (−2.31, −0.75)); virtual reality exercise (MD = −1.45, 95% CI = (−2.45, −0.45)); Pilates (MD = −1.36, 95% CI = (−1.83, −0.88)); and resistance exercise (MD = −0.79, 95% CI = (−1.55, −0.02)) were superior to the control group in terms of reducing TUG time (Table 4). Aquatic exercise achieved the number one SUCRA probability ranking in terms of reducing the TUG time (SUCRA: 78.8%). Details are shown in Figure 3. The *p*-values for the heterogeneity tests can be found in Appendix A.

### 3.5. Publication Bias

Looking at the parallelism of the horizontal line in the funnel plot to the *x*-axis, we concluded that there was no publication bias among the original studies that affected the NMA (Figure 4) [54].

## 4. Discussion

Exercise therapy is shown to be effective as a rehabilitation measure to improve physical function and to promote neuroplasticity [9]. The combined effect of improved physical function and mental rehabilitation helps to reduce the risk of falls for people with MS. The results of previous meta-analyses show that different types of exercise therapy result in good improvements in the physical and mental functional abilities of people with MS compared to traditional rehabilitation measures. Harrison et al. ranked the different exercise interventions in their study to find the best exercise intervention for improving the mental health of people with MS [13]. However, controversy remains in the review regarding which exercises are most the effective in improving the physical function of people with MS. Our study explored the effects of different exercises on the physical functional abilities of people with MS for the first time. The results of the meta-analysis show that yoga is the best intervention to improve dynamic and static balance for people with MS and that aquatic exercise is the best intervention to improve the functional walking ability of people with MS, based on improvements in the BBS Balance Scale and the TUG test, respectively.

The BBS test is a comprehensive functional test that reflects the ability of MS patients to actively shift their center of gravity by examining their dynamic and static balance in a sitting or standing position; its results are accurate and acceptable [55]. The results of this study show that yoga is superior to other interventions in terms of improving patients’ dynamic and static balance and postural control (MD = 5.50, 95% CI = (2.55, 8.45)). The mechanism by which yoga improves BBS can be explained in several ways: (1) Yoga practice emphasizes the control of the nervous system over the muscles through less intense stretching-type activities, increasing the unconscious specific response of the muscles to dynamic joint stability signals and emphasizing the control of the core muscles throughout the whole trunk during the activity [56]. (2) Yoga training involves the training of joint function, which allows for the normal degree of motion in the joints to be maintained and deformed postures to be corrected through active or passive stretching to relieve abnormal tension in the joint capsule [57]. (3) Some yoga postures are performed with the eyes closed, emphasizing increased attention to the other sensory organs, particularly the vestibular organs [37].

In addition to dynamic static balance, improvements in physical function are also associated with increased functional walking ability, and the TUG test is often used to test the functional walking ability of people with MS due to its ease of administration and the sensitivity of the test [58]. Our findings suggest that aquatic exercise is superior to other interventions in improving functional walking ability. The mechanism by which aquatic exercise improves TUG can be explained in several ways: (1) The presence of water resistance allows the patient to perform exercises more slowly than land-based exercises, resulting in an increased weight-bearing time on the lower limbs; in addition, the patient’s torso is subjected to a certain amount of water pressure in the aquatic environment [59] which has a similar effect on the skeletal muscles as blood flow restriction training [60] (a training modality that was shown to have a significant effect on increasing muscle strength) [61]. (2) When exercising in the water, hydrostatic force causes the blood and lymphatic fluid to move up the torso, and combined with the gravitational offload and hydrostatic effect of the water, it increases the amount of blood circulating from the periphery to the center, resulting in an increase in the end-diastolic volume of the heart and thus an increase in cardiac output [62]. This increase in blood volume first reaches the brain and muscle tissue and is accompanied by an increase in serum brain-derived neurotrophic factor (BDNF), an anti-inflammatory factor that is particularly important for brain and muscle recovery [63]. (3) The period during and after aquatic exercise causes the body to reduce sympathetic activity and improves sympathetic–parasympathetic balance by increasing vagal tone [64].

The rehabilitative effects of both exercises in their respective domains are very useful, and we therefore recommend that MS patients prioritize both exercises when rehabilitating their physical function. In addition, we hypothesize that yoga and aquatic training can be alternated during a full rehabilitation cycle when conditions permit, but due to the lack of direct clinical evidence, we maintain a wait-and-see attitude towards this combined intervention in hopes that further experiments will prove or disprove our assumptions.

## 5. Strengths and Limitations

Our study has several strengths and weaknesses. Firstly, our study focused on finding the most effective exercise intervention for treating the physical function of people with MS among a variety of exercise interventions; this was not addressed in previous studies. Secondly, our study only included randomized controlled trials, which is the “gold standard” in the field of clinical research.

Admittedly, there are certain limitations in both our study and the original studies included. Heterogeneity between each of the original studies is inevitable (e.g., the ratio between male and female participants; the original studies are from different regions), and this heterogeneity can affect the scientific validity of the network meta-analysis to some extent. In addition, we did not include tests used to evaluate the physical function of MS patients in this study because there are too few original studies regarding these tests. However, we remain hopeful that more original studies will expand on the results of this study in the future in order to update and provide more solid clinical evidence-based recommendations.

In our study, readers should interpret the results with caution because of the small number of studies included and the limited head-to-head direct comparative evidence for some interventions. Our study highlights the need for the further expansion of relevant studies and their timeliness.

## 6. Conclusions

Based on studies using the Berg Balance Scale and the Timed-Up-and-Go test, we suggest that the exercise interventions discussed in this paper, compared with conventional care, all had an effect on improving the dynamic and static balance and the functional walking ability of patients with MS. However, yoga training, virtual reality training and aerobic training were more effective in improving dynamic and static balance; aquatic exercise, aerobic training and virtual reality training were more effective in improving functional walking ability.

## Figures and Tables

**Figure 1 ijerph-19-07175-f001:**
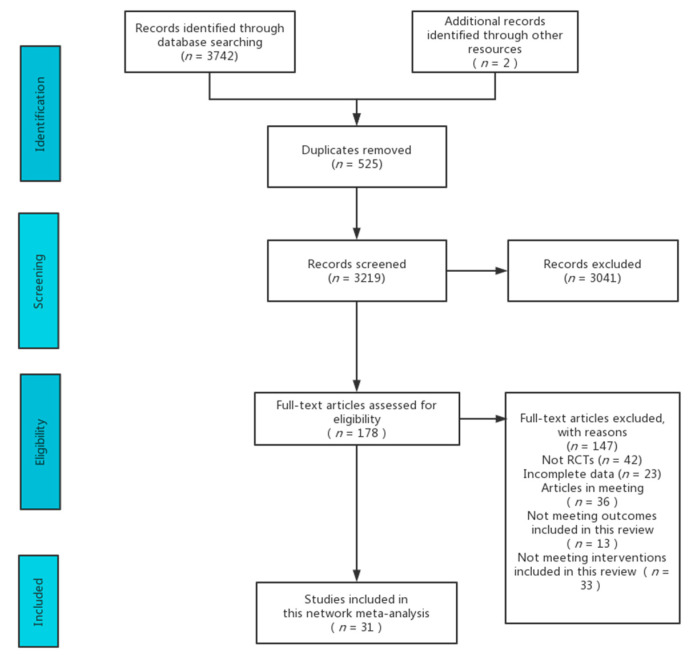
Flow diagram of literature selection.

**Figure 2 ijerph-19-07175-f002:**
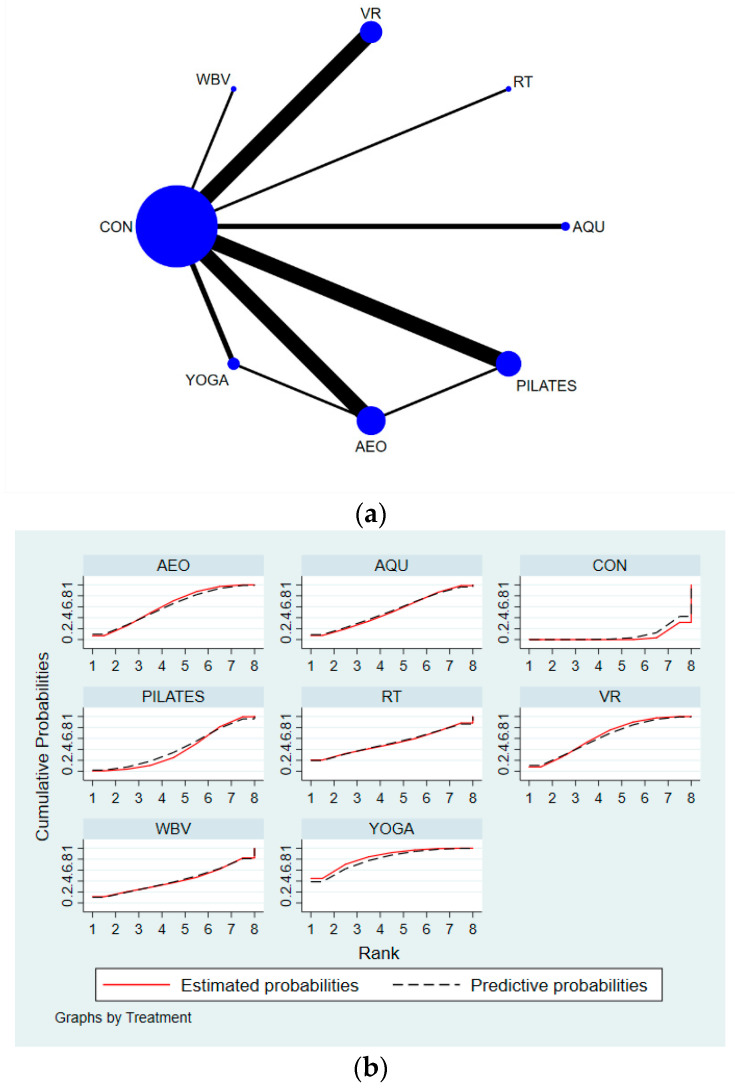
Specific details regarding the BBS’ network meta-analysis: (**a**) NMA plot for BBS and (**b**) SUCRA plot for BBS. AQU: aquatic exercise; AEO: aerobic exercise; YOGA: yoga; PILATES: Pilates; VR: virtual reality exercise; WBV: whole-body vibration exercise; RT: resistance exercise.

**Figure 3 ijerph-19-07175-f003:**
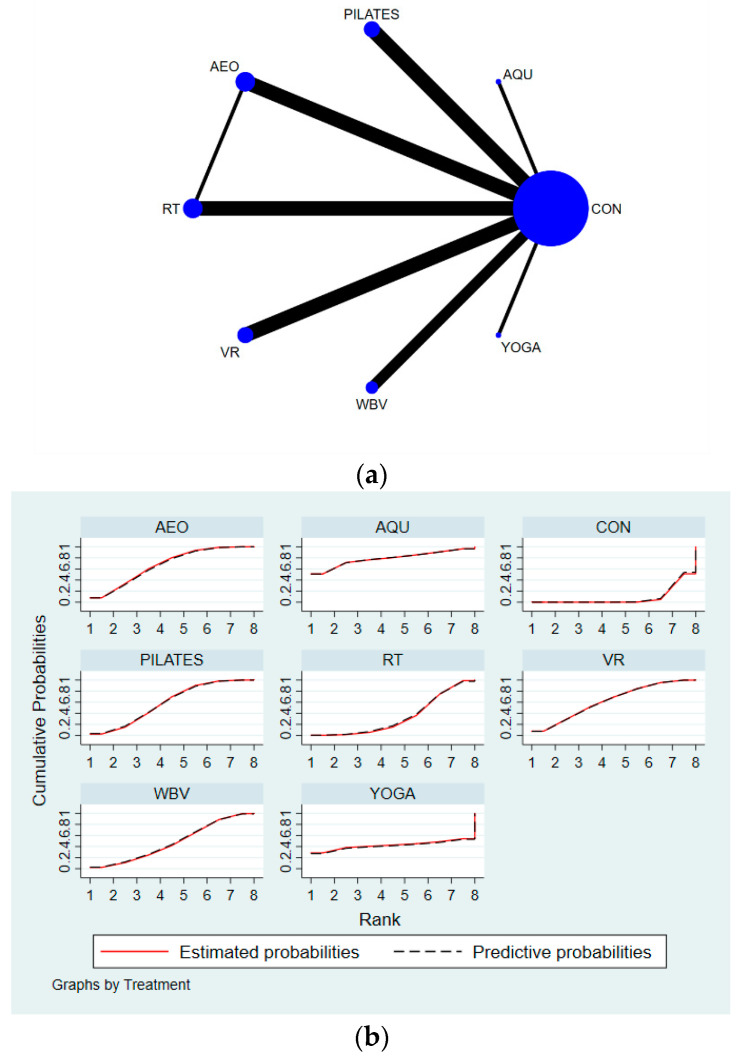
Specific details regarding the network meta-analysis for TUG: (**a**) NMA plot for TUG and (**b**) SUCRA plot for TUG.

**Figure 4 ijerph-19-07175-f004:**
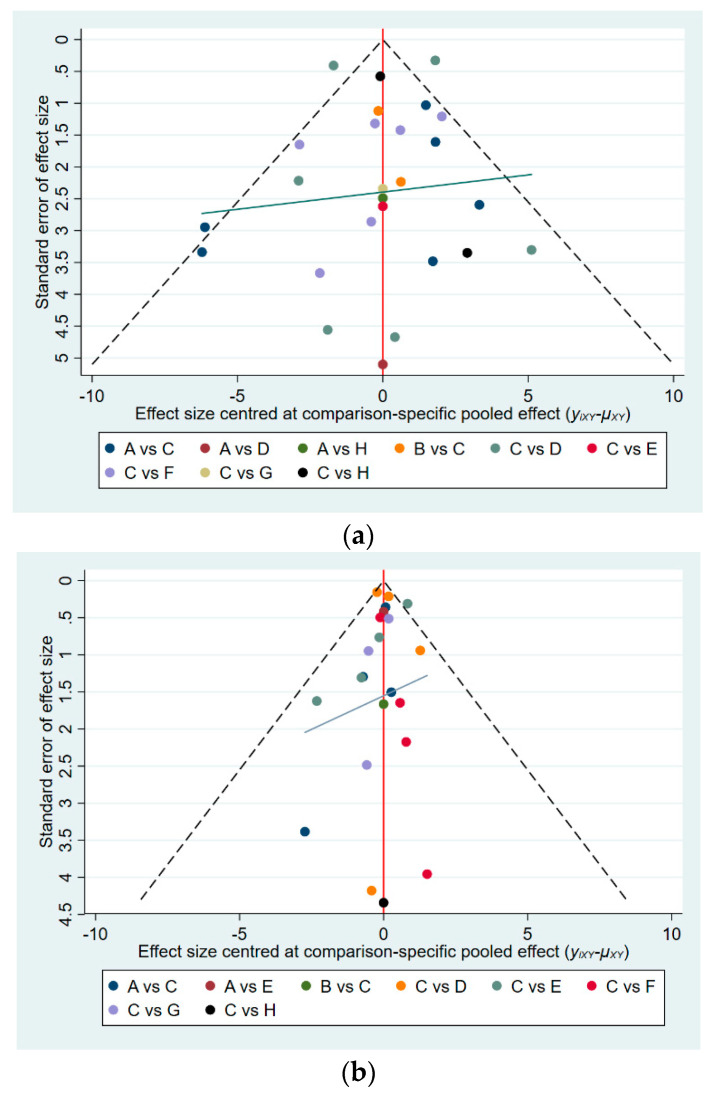
Funnel plot to test for publication bias: (**a**) funnel plot for BBS and (**b**) funnel plot for TUG. A: aerobic training, B: aquatic training, C: control group, D: Pilates, E: resistance training, F: virtual reality training, G: whole-body vibration training, H: yoga.

**Table 1 ijerph-19-07175-t001:** Search strategy on Pubmed.

#1	Search “Multiple Sclerosis” [MeSh]
#2	Search (Multiple Sclerosis [Title/Abstract]) OR (MS [Title/Abstract]) OR (Relapsing-remitting Multiple Sclerosis [Title/Abstract]) OR (RRMS [Title/Abstract]) OR (Multilocular Sclerosis [Title/Abstract])
#3	Search #1 OR #2
#4	Search “Exercise” [MeSh]
#5	Search (exercise [Title/Abstract]) OR (exercise intervention [Title/Abstract]) OR (exercise training [Title/Abstract]) OR (training [Title/Abstract]) OR (physical training [Title/Abstract]) OR (physical exercise [Title/Abstract]) OR (sports training [Title/Abstract]) OR (nurse intervention [Title/Abstract]
#6	Search #4 OR #5
#7	Search #3 AND #6

**Table 2 ijerph-19-07175-t002:** Details of the studies included in the network meta-analysis.

Author	Country	Year	Age (Mean + SD)	Total/Man/Woman	Intervention 1/Length (Weeks)/Frequency (Weeks)/Duration (Minutes)	Intervention 2/Length (Weeks)/Frequency (Weeks)/Duration (Minutes)	Control Group	Outcome
Ahmadi	Iran	2010	T: 36.8 (9.17) C: 36.7 (9.32)	T: 10/NA/NA C: 10/NA/NA	Aerobic training 8 weeks 3 times a week 30 min	NA	No exercise	BBS
Ahmadi	Iran	2013	T1: 36.8 (9.17) T2: 32.27 (8.68) C: 36.7 (9.32)	T1: 10/NA/NA T2: 11/NA/NA C: 10/NA/NA	Aerobic training 8 weeks 3 times a week 30 min	Yoga training 8 weeks 3 times a week 60 min	No exercise	BBS
Gervasoni	Italy	2014	T: 49.6 (9.4) C: 45.7 (8.9)	T: 15/NA/NA C: 15/NA/NA	Aerobic training 2 weeks 6 times a week 45 min	NA	No exercise	BBS
Straudi	Italy	2015	T: 52.26 (11.11) C: 54.12 (11.44)	T: 27/10/17 C: 25/8/17	Aerobic training 2 weeks 2 times a week 30 min	NA	Usual care	BBS, TUG
Tollar	Hungary	2019	T: 48.1 (5.65) C: 44.4 (6.76)	T: 14/1/13 C: 12/1/11	Aerobic training 5 weeks 5 times a week 40 min		No exercise	BBS
Cakt	Turkey	2010	T1: 36.4 (10.5) T2: 35.5 (10.9) C: 43 (10.2)	T1: 14/5/9 T2: 9/3/6 C: 10/2/8	Aerobic training 8 weeks 2 times a week 20 min	Resistance training 8 weeks 2 times a week 20 min	No exercise	TUG
Orban	USA	2019	T: 44.7 (9.4) C: 48.7 (8.4)	T: 10/1/9 C: 7/2/5	Aerobic training 8 weeks 4 times a week 30 min		No exercise	TUG
Straudi	Italy	2013	T: 49.92 (7.51) C: 55.25 (13.82)	T: 12/5/7 C: 12/2/10	Aerobic training 6 weeks 2 times a week 30 min		No exercise	TUG
Asvar	Iran	2020	T: 32.1 (13) C: 33.9 (6)	T: 15/0/15 C: 15/0/15	Pilates training 8 weeks 3 times a week 60 min		No exercise	BBS
Gheitasi	Iran	2020	T: 30.6 (5.27) C: 32.1 (6.3)	T: 15/15/0 C: 15/15/0	Pilates training 12 weeks 3 times a week 50 min		No exercise	BBS, TUG
Gunduz	Turkey	2014	T: 36 (29-40) C:36 (27-45)	T: 18/NA/NA C: 8/NA/NA	Pilates training 8 weeks 2 times a week 60 min		Usual care	BBS, TUG
Karlon	Israel	2016	T: 42.9 (7.2) C: 44.3 (6.6)	T: 23/8/14 C: 22/8/15	Pilates training 12 weeks 1 time a week 30 min		Usual care	BBS, TUG
Kara	Turkey	2017	T1: 49.77 (8.95) T2: 43.03 (10.26) C: 44.42 (5.98)	T1: 9/NA/NA T2: 26/NA/NA C: 21/NA/NA	Pilates training 8 weeks 2 times a week 45 min	Aerobic training 8 weeks 2 times a week 45 min	No exercise	BBS
Kucuk	Turkey	2016	T: 47.2 (9.5) C: 49.7 (8.9)	T: 11/NA/NA C: 9/NA/NA	Pilates training 8 weeks 2 times a week 60 min		Usual care	BBS
Zuhal	Turkey	2019	T: 42.5 (6.76) C: 48.24 (11.79)	T: 16/NA/NA C: 17/NA/NA	Pilates training 8 weeks 3 times a week 60 min		Usual care	TUG
Gerson	Brazil	2016	T: 46 (8) C: 45 (9)	T: 6/NA/NA C: 6/NA/NA	Yoga training 24 weeks 2 times a week 60 min		No exercise	BBS
Yazgan	Turkey	2020	T: 47.76 (10.53) C: 40.66 (8.82)	T: 15/2/13 C: 15/2/13	VR training (Nintendo® Wii®) 8 weeks 2 times a week 60 min		No exercise	BBS, TUG
Khalil	Jordan	2019	T: 39.88 (12.75) C: 34.87 (8.98)	T: 16/4/12 C: 16/6/10	VR training (VR scenarios) 6 weeks 3 times a week 30 min		Usual care	BBS, TUG
Brichetto	Italy	2013	NA	T: 18/NA/NA C: 18/NA/NA	VR training (Nintendo® Wii®) 4 weeks 3 times a week 60 min		Usual care	BBS
Lozana	Spain	2014	T: 48.33 (10.82) C: 40.6 (9.24)	T: 6/3/3 C: 5/4/1	VR training (Kinect games) 10 weeks 1 time a week 60 min		No exercise	BBS, TUG
Molhemi	Iran	2020	T: 36.8 (8.4) C: 41.6 (8.4)	T: 19/7/12 C: 20/8/12	VR training (Kinect games) 6 weeks 3 times a week 35 min		Usual care	BBS, TUG
Tollar	Hungary	2019	T: 48.2 (5.48) C: 44.4 (6.76)	T: 14/2/12 C: 12/1/11	VR training (Nintendo® Wii®) 5 weeks 5 times a week 40 min		No exercise	BBS
Aidar	Brazil	2018	T: 41.3 (7.3) C: 43.6 (7.6)	T: 13/4/9 C: 13/5/8	Aquatic training 12 weeks 3 times a week 45–60 min		No exercise	BBS, TUG
Kargarfard	Iran	2017	T: 36.5 (9) C: 36.2 (7.4)	T: 17/NA/NA C: 15/NA/NA	Aquatic training 8 weeks 3 times a week 60 min		No exercise	BBS
Aidar	Brazil	2018	T: 42.8 (8) C: 43.6 (7.7)	T: 11/4/7 C: 12/4/8	Resistance training 12 weeks 3 times a week 45–60 min		No exercise	BBS, TUG
Moradi	Iran	2015	T: 34.38 (11.07) C: 33.13 (7.08)	T: 8/NA/NA C: 10/NA/NA	Resistance training 8 weeks 6 times a week 30 min		No exercise	TUG
Moghadasi	Iran	2020	T: 37.62 (4.58) C: 34.72 (5.01)	T: 16/NA/NA C: 11/NA/NA	Resistance training 8 weeks 3 times a week 30 min		No exercise	TUG
Alguacil	Spain	2012	T: 44 (20) C: 43 (17)	T: 15/7/8 C: 17/9/8	Whole-body vibration training 5 days 1 time a day 10 min		No exercise	BBS, TUG
Broekmans	Belgium	2010	T: 46.1 (2.1) C: 49.7 (3.3)	T: 11/7/4 C: 14//11/3	Whole-body vibration training 20 weeks 2 times a week 10 min		No exercise	TUG
Schuhfried	Austria	2005	T: 49.3 (13.3) C: 46 (12.7)	T: 6/1/5 C: 6/2/4	Whole-body vibration training 2 weeks 4 times a week 15 min		Usual care	TUG
Young	USA	2018	T: 48.35 (9.95) C: 47.29 (10.33)	T: 26/6/20 C: 29/4/24	Yoga training 8 weeks 3 times a week 40 min		No exercise	TUG

T: experimental group; C: control group; BBS: Berg Balance Scale; TUG: Timed-Up-and-Go score; Freq: frequency; NA: no mention in the original article.

**Table 3 ijerph-19-07175-t003:** League table for BBS.

YOGA	VR	AEO	AQU	RT	PILATES	WBV	CON
YOGA	−1.38 (−4.92, 2.16)	−1.49 (−4.92, 1.93)	−2.07 (−6.42, 2.28)	−2.06 (−8.83, 4.70)	−2.80 (−6.35, 0.76)	−3.05 (−9.42, 3.31)	−5.50 (−8.45, −2.55)
1.38 (−2.16, 4.92)	VR	−0.11 (−3.09, 2.86)	−0.69 (−4.44, 3.07)	−0.68 (−7.08, 5.72)	−1.42 (−4.22, 1.39)	−1.67 (−7.65, 4.30)	−4.12 (−6.09, −2.15)
1.49 (−1.93, 4.92)	0.11 (−2.86, 3.09)	AEO	−0.58 (−4.46, 3.31)	−0.57 (−7.04, 5.90)	−1.30 (−4.24, 1.63)	−1.56 (−7.61, 4.49)	−4.01 (−6.20, −1.81)
2.07 (−2.28, 6.42)	0.69 (−3.07, 4.44)	0.58 (−3.31, 4.46)	AQU	0.01 (−6.87, 6.89)	−0.73 (−4.50, 3.04)	−0.98 (−7.47, 5.50)	−3.43 (−6.63, −0.23)
2.06 (−4.70, 8.83)	0.68 (−5.72, 7.08)	0.57 (−5.90, 7.04)	−0.01 (−6.89, 6.87)	RT	−0.74 (−7.14, 5.67)	−0.99 (−9.29, 7.31)	−3.44 (−9.53, 2.65)
2.80 (−0.76, 6.35)	1.42 (−1.39, 4.22)	1.30 (−1.63, 4.24)	0.73 (−3.04, 4.50)	0.74 (−5.67, 7.14)	PILATES	−0.25 (−6.24, 5.73)	−2.70 (−4.69, −0.71)
3.05 (−3.31, 9.42)	1.67 (−4.30, 7.65)	1.56 (−4.49, 7.61)	0.98 (−5.50, 7.47)	0.99 (−7.31, 9.29)	0.25 (−5.73, 6.24)	WBV	−2.45 (−8.09, 3.19)
5.50 (2.55, 8.45)	4.12 (2.15, 6.09)	4.01 (1.81, 6.20)	3.43 (0.23, 6.63)	3.44 (−2.65, 9.53)	2.70 (0.71, 4.69)	2.45 (−3.19, 8.09)	CON

AQU: aquatic exercise; AEO: aerobic exercise; YOGA: yoga; PILATES: Pilates; VR: virtual reality exercise; WBV: whole-body vibration exercise; RT: resistance exercise; CON: control group.

**Table 4 ijerph-19-07175-t004:** League table for TUG.

AQU	AEO	VR	PILATES	WBV	YOGA	RT	CON
AQU	1.05 (−2.34, 4.44)	1.13 (−2.32, 4.58)	1.22 (−2.11, 4.56)	1.48 (−1.95, 4.92)	2.28 (−6.86, 11.42)	1.79 (−1.59, 5.18)	2.58 (−0.72, 5.88)
−1.05 (−4.44, 2.34)	AEO	0.08 (−1.20, 1.36)	0.17 (−0.76, 1.10)	0.43 (−0.80, 1.66)	1.23 (−7.33, 9.79)	0.74 (−0.19, 1.67)	1.53 (0.75, 2.31)
−1.13 (−4.58, 2.32)	−0.08 (−1.36, 1.20)	VR	0.09 (−1.00, 1.19)	0.35 (−1.04, 1.74)	1.15 (−7.43, 9.73)	0.66 (−0.63, 1.95)	1.45 (0.45, 2.45)
−1.22 (−4.56, 2.11)	−0.17 (−1.10, 0.76)	−0.09 (−1.19, 1.00)	PILATES	0.26 (−0.82, 1.34)	1.06 (−7.48, 9.59)	0.57 (−0.41, 1.55)	1.36 (0.88, 1.83)
−1.48 (−4.92, 1.95)	−0.43 (−1.66, 0.80)	−0.35 (−1.74, 1.04)	−0.26 (−1.34, 0.82)	WBV	0.80 (−7.78, 9.38)	0.31 (−0.88, 1.51)	1.10 (0.15, 2.05)
−2.28 (−11.42, 6.86)	−1.23 (−9.79, 7.33)	−1.15 (−9.73, 7.43)	−1.06 (−9.59, 7.48)	−0.80 (−9.38, 7.78)	YOGA	−0.49 (−9.05, 8.07)	0.30 (−8.22, 8.82)
−1.79 (−5.18, 1.59)	−0.74 (−1.67, 0.19)	−0.66 (−1.95, 0.63)	−0.57 (−1.55, 0.41)	−0.31 (−1.51, 0.88)	0.49 (−8.07, 9.05)	RT	0.79 (0.02, 1.55)
−2.58 (−5.88, −0.72)	−1.53 (−2.31, −0.75)	−1.45 (−2.45, −0.45)	−1.36 (−1.83, −0.88)	−1.10 (−2.05, −0.15)	−0.30 (−8.82, 8.22)	−0.79 (−1.55, −0.02)	CON

AQU: aquatic exercise; AEO: aerobic exercise; YOGA: yoga; PILATES: Pilates; VR: virtual reality exercise; WBV: whole-body vibration exercise; RT: resistance exercise; CON: control group.

## Data Availability

The data that support the findings of the study are available from the first author upon reasonable request.

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
