# Peer review of "Effects of Different Exercise Therapies on Balance Function and Functional Walking Ability in Multiple Sclerosis Disease Patients—A Network Meta-Analysis of Randomized Controlled Trials"

_ijerph, 2022, doi:10.3390/ijerph19127175_

Round 1

Reviewer 1 Report

Dear authors, Thank you for the opportunity to review your manuscript. It is interesting to evaluate the impact of the different rehabilitation approaches for people with multiple sclerosis

17 this is not design, but materials and methods.

24 is not a sample size, but "we included n RTCs with n patients"

Mentioning software in the abstract is superfluous.

The presented MD refers to what? because the ranking orders the interventions by placing the network ES above the diagonal .. and below, the pairwise ES. So, considering the use of STATA, have you thought about reporting each SUCRA scores?

32 rehabilitation

36 is not autoimmune .. follow the bibliographic reference ref: https://link.springer.com/article/10.1007/s11910-021-01110-5

43 repetition

54 the good news is inappropriate

155 I suggest adding the description of net league table and references “Thus in a ranking table, Treatments were ranked from best to worst along the leading diagonal. Above the leading diagonal were estimates from pairwise meta-analyses, below the leading diagonal were estimates from network meta-analyses.”
Ref: http://dx.doi.org/10.1016/j.ctcp.2020.101260

156 actually the P score is not mentioned in the results, so I would suggest describing only the SUCRA

I would recommend with the characteristics of the studies a list / table on the type of intervention .. because VR training includes several interventions .. https://doi.org/10.1016/j.msard.2021.103128

233 For the presentation of these results I would like to see a forest with each intervention versus the network control

Both Figure C… do not seem to me that the area under the curve transmits 80% for AQU.. I would like to know all the SUCRA scores

analyzing the curves, if we consider the SUCRA scores ... I don't think there are high probability values, but above all I think the interventions are very little detached from each other, so much so that there are several therapies that have excellent and similar results and not just one at the top of the chart ..

This is essential to conclusively define whether one intervention is really better than another and in what size

So I recommend a major revision

Author Response

Dear Reviewer,

Thank you very much for your scientifically sound and rigorous advice, we have revised our thesis in line with your suggestions. Please forgive our shortcomings in the use of stata software, but the references you provided us with as an expert in the field and your sincere advice played a crucial role in the revision process, which made our paper look more scientific and easier to read and understand, thank you very much.Here is our response to your suggestion.

Response to Reviewer 1 Comments

For your responses, we present them in the manuscript with annotations in red.

Point 1: 17 this is not design, but materials and methods.

Response 1: We changed "Design" to "Materials and methods".

Point 2: 24 is not a sample size, but "we included n RCTs with n patients"

Response 2: We changed "sample size" to "We included 31 RCTs with 904 patients".

Point 3: Mentioning software in the abstract is superfluous. The presented MD refers to what? because the ranking orders the interventions by placing the network ES above the diagonal .. and below, the pairwise ES. So, considering the use of STATA, have you thought about reporting each SUCRA scores?

Response 3:

  • Following your suggestion, we have removed the description of the software from the original text.
  • We will highlight in the abstract section that we are using the SUCRA ranking to determine the likely effects of different exercise interventions.

Point 4: 32 rehabilitation

Response 4: We changed "exercise therapy" to "rehabilitation".

Point 5: 36 is not autoimmune ..

Response 5: In conjunction with another reviewer's comments, we decided to delete 36 to 38, and when we looked at the sentence again, we felt that it lacked not only scientific validity but also innovation.

Point 6: 43 repetition

Response 6: We have removed the duplicates.

Point 7: 54 the good news is inappropriate

Response 7: We have removed the “the good news is that”.

Point 8: 155 I suggest adding the description of net league table and references “Thus in a ranking table, Treatments were ranked from best to worst along the leading diagonal. Above the leading diagonal were estimates from pairwise meta-analyses, below the leading diagonal were estimates from network meta-analyses.” and 156 actually the P score is not mentioned in the results, so I would suggest describing only the SUCRA

Response 8: We have taken your suggestion into consideration and replaced the original with the statement you recommended to us and quoted it, which makes our article look more scientific.

Point 9: I would recommend with the characteristics of the studies a list / table on the type of intervention .. because VR training includes several interventions ..

Response 9 : Thanks to your advice, we have combined the suggestions of another expert and modified the table. We have made an addition to Table2 to differentiate the different VR.

Point 10: 233 For the presentation of these results I would like to see a forest with each intervention versus the network control

Response 10: It may be that the difference in the code in the stata software that you and I use makes the forest diagram different. We have uploaded a different kind of forest diagram, and we don't know if it is the ideal style you have in mind. We have also decided that the forest diagram is not as important as SUCRA for network meta-analysis, so we have removed the forest diagram from the manuscript and will add it to the manuscript if you think it is important.

 Figure 1. Forest diagram for BBS(They will be presented in the annotations to the manuscript.)

Figure 2. Forest diagram for TUG(They will be presented in the annotations to the manuscript.)

Point 11: Both Figure C… do not seem to me that the area under the curve transmits 80% for AQU.. I would like to know all the SUCRA scores

analyzing the curves, if we consider the SUCRA scores ... I don't think there are high probability values, but above all I think the interventions are very little detached from each other, so much so that there are several therapies that have excellent and similar results and not just one at the top of the chart ..

This is essential to conclusively define whether one intervention is really better than another and in what size

Response 11:

  • In terms of the interpretation of the SUCRA problem, we believe that it may be due to a discrepancy with the code in the stata software you used, which we used before, the code "sucra prob*, labels(A B C D E F) rankog".
  • Due to your expertise in this field, we have re-read the references you sent us and have used the relevant codes to calculate the area of SUCRA and its corresponding graph. We have modified the recalculated value and the graph it presents in the original text. Here are twotables of all the SUCRA values you would like to know about.
  • Perhaps our understanding of these exercise interventions is not as deep as yours, so we have focused too much on the one-sided top rankings, so we have rewritten the conclusions in the abstract section based on your suggestions.

Table 1 SUCRA for BBS

Treatm~t

SUCRA

PrBest

MeanRank

AEO

60.6

9.7

3.8

AQU

52.9

8.9

4.3

CON

8.5

0.1

7.4

PILATES

41.8

2.0

5.1

RT

52.3

19.7

4.3

VR

62.7

10.5

3.6

WBV

41.5

10.4

5.1

YOGA

79.7

38.7

2.4

Table 2 SUCRA for TUG

Treatm~t

SUCRA

PrBest

MeanRank

AEO

66.8

7.8

3.3

AQU

78.8

50.9

2.5

CON

8.7

0.1

7.4

PILATES

59.6

3.3

3.8

RT

33.9

0.4

5.6

VR

62.4

7.5

3.6

WBV

48.3

2.8

4.6

YOGA

41.5

27.4

5.1

Author Response

Dear Reviewer,

We were pleasantly surprised by your suggestions, it was so careful and conscientious and you took into account a lot of details. These suggestions made the whole article look more scientific and flowing and easier for the readers to read. Thank you for your dedication to our paper. We used MDPI Magazine's English language touch-up service at your suggestion. Here is our response to your suggestion.

Response to Reviewer 2 Comments

For your responses, we present them in the manuscript with annotations in blue.

Overall Impression: 

Point: In this manuscript by Hao, Zhang and Chen, the authors aim to evaluate different exercise therapies (aquatic, aerobic, virtual reality, whole body vibration and resistance exercises, yoga and Pilates). I have accepted to review this article as I believe that the idea is worthy of publication. However, the authors are not critical enough of the methodology they are using. While Network meta-analysis (NMA) is a “state-of-the-art methods for comparing the effect of multiple interventions on a disease using direct or indirect comparisons and for estimating the rank order of each intervention,” it is not a technique that has no limitations. I highly recommend that the authors discuss the limitations and potential pitfalls of the technique for a balanced publication. Reviewing the article cited by the reference (article # 14) on NMA, the article emphasizes the importance of using caution in interpreting the data. “The clinical utility of interventions can be better understood if the analysis considers both effectiveness and safety outcomes.” Clearly this meta-analysis fails to consider the safety of the interventions. If the authors deem that the interventions are 100% safe, it is perfectly acceptable to say that.

Response: Thank you very much for your endorsement of our article's point of intention and methodological suggestions. As you say, the NMA, while novel enough, is not without its drawbacks and we have added a discussion of its limitations in the relevant section. Secondly, as we believe that exercise interventions are safe when monitored by each researcher, safety is not discussed in this article, but we would still like more studies to report on anything in terms of safety to standardise future experimental studies and we would actively consider an NMA analysis if the number of studies on safety reaches a certain level.

Introduction

Point 1: I highly recommend deleting 36 to 38 as the idea is cliché.

Response 1: We have removed this sentence in response to your suggestion.

Point 2: The prevalence of MS worldwide was 2.5 million when the prevalence in the United States used to be 400,000. We now know that the prevalence in the United States is about 1 million and therefore the 2.5 million worldwide is an old estimate. I recommend the authors to find an updated reference.

Response 2: Based on your comments, and so as not to mislead the reader, we think it would be more appropriate to change the sentence to read "As of 2017, the number of people with multiple sclerosis is 2.5 million worldwide, and this number is increasing." Perhaps it would be more appropriate.

Point 3: Lines 42 to 44 are duplicate sentences.

Response 3: We have removed the duplicates.

Point 4: I highly recommend that the authors switch “MS patients” to patients with MS or people with MS (pwMS). This last one specifically is becoming somewhat of a standard in more recent publications.

Response 4: We have replaced the relevant word from "MS patients" to "pwMS" throughout the text.

Point 5: Line 58: what did the authors mean by “produces effects”? Did they mean that exercise is beneficial?

Response 5: Thanks to your care and rigour, we have changed "produces effects" to "produces beneficial effects ".

Point 6: Line 67 to 69 and 69 to 72 are duplicate.

Response 6: We have removed the duplicates.

Materials and methods

Point 1: Line 80: suggest switching “researchers” by “authors.”

Response 1: We have changed "researchers" to "authors ".

Point 2: Inclusion criteria: I suggest that the authors define “routine care and rehabilitation only” in line 93. 

Response 2: We define "routine care and rehabilitation only" as "no other types of exercise interventions, just the more popular and commonly used balance rehabilitation exercises; no training, just daily living care".

Point 3: Outcomes: why did the author choose the Berg balance scale and the timed up and go scores? Please justify and explain how does the BBS or TUG compare or contrast to other scales.

Response 3: BBS and TUG, which are popular with clinical practitioners because of their comprehensiveness, sensitivity and simplicity.

Point 4: Study selection: a. I suggest switching “to researchers” to “the authors.” b. I suggest switching the word “literature” in line 113 to “articles” or “papers.”

Response 4: We have changed "researchers" to "authors " and literature to papers.

Point 5: Data extraction: please clarify what was the “seven-item, standardized and preselected data extraction?”

Response 5: We looked back at the sentence again and decided that our expression might cause confusion to the reader, so we replaced the sentence with "A table with seven sections was used to extract detailed data from the included papers.

Point 6: Risk of bias of individuals studies: a. Regarding stratification of risk of bias, did the authors use a previously reported method or their own? If the method was their own, it should be clarified. b. The idea in lines 161 to 164 is unclear and the sentence is incomplete.

Response 6:

  • We indicate at the beginning of this section that this approach is derived from previous research (Cochrane Handbook version 5.1.0 tool).
  • We have combined your suggestions with those of another reviewer and have rewritten the section. Rewrite the sentence as "Thus in a ranking table, Treatments were ranked from best to worst along the leading diagonal. Above the leading diagonal were estimates from pairwise meta-analyses, below the leading diagonal were estimates from network meta-analyses. ".

Results

Point 1: I suggest switching “documents” to “articles” in line 172.

Response 1: We have changed "documents" to "articles ".

Figures and tables

Point 1: Figure 1: In the last two boxes, the authors mean qualitative and quantitative respectively?

Response 1: The issue you mention is because the process in this diagram was created by us based on previous research. The issue you raise is critical because it can cause misunderstandings among readers, so we have simplified the process and recreated the flowchart.

Point 2: Table2: a suggest that the authors use the words “men” and “women” rather than “male” and “female” if this is a “human” study. b. In columns “Intervention 1” and Intervention 2, rather than repeating “length of intervention”, “frequency” and “duration” in every row, I suggest writing it once in the heading such as “Intervention 1/ length (weeks)/ frequency (weeks) /Duration (minutes). This is easier for following up and comparison of the information per study. c. There is no separating line between the “Gerson” and “Yazgan” studies

Response 2: 

  • We have changed male and femaleto man and woman.
  • Thank you for your suggestion, the whole table does look a lot clearer after we have made the changes.
  • Thank you for your care, we have made the change.

Point 3: Figure 2a and 3a: I suggest that the authors add the significance of the circles and lines size in the legend, the way they explained it in the text.

Response 3: We have explained the picture in this section of 2.8: the different nodes represent different interventions; the size of the nodes represents the sample size of the patients; the thickness of the connecting lines represents the sample size of the included literature.

Point 4: Please provide a legend and explain why some results are in red and others are not.

Response 4: We apologise for the inconvenience caused by our carelessness in reading this, we have made the colours a uniform style.

Point 5: Figure 4: It’s unclear what the letters A to H refer to.

Response 5: This image was created by stata software and due to the limited space of the image, we have used letters for each intervention. After modification, we have explained each letter at the bottom of the picture. A : Aerobic training, B : Aquatic training, C : Control group, D : Pilates training, E : Resistance training, F : Virtual reality exercises, G : Whole body vibration training, H : Yoga training

Discussion

Point 1: Regarding the BBS: the authors justify why they used the scale, which is great, but did not explain how it supersedes other scales. The same argument applies to the use of the TUG test.

Response 1: BBS and TUG, which are popular with clinical practitioners because of their comprehensiveness, sensitivity and simplicity.

Point 2: Lines 294 to 295 is incomplete.

Response 2: We have completed the whole sentence to make it more accurate.

Point 3: Suggest rephrasing 322 to 324 to “The period during and after water exercise causes the body to reduce sympathetic activity and improves sympathetic – parasympathetic balance by increasing vagal tone.

Response 3: We have made changes to the original text in response to your suggestions.

Point 4: Please delete “which” in line 324.

Response 4: We have made changes to the original text in response to your suggestions.

Point 5: It is unclear what the authors are trying to say from lines 330 to 333 and from line 348 to 352.

Response 5: Our study has several strengths and weaknesses. Firstly, our study focused on finding the most effective exercise intervention for treating physical function in people with MS among a variety of exercise interventions. This has not been addressed in previous studies, and secondly, our study only included randomised controlled trials, which is the 'gold standard' in the field of clinical research.

Strengths and limitations

Point 1: The idea in lines 348 to 351 is unclear.

Response 1: Our study has several strengths and weaknesses. Firstly, our study focused on finding the most effective exercise intervention for treating physical function in people with MS among a variety of exercise interventions. This has not been addressed in previous studies, and secondly, our study only included randomised controlled trials, which is the 'gold standard' in the field of clinical research.

Point 2: Lines 355-356: please rephrase for the sake of clarity.

Response 2: We have made changes to the original text in response to your suggestions. Admittedly, there are certain limitations in both our study and the original studies included. Heterogeneity between each of the original studies is inevitable (e.g. the ratio between male and female participants; the original studies are from different regions) and this heterogeneity can affect the scientific validity of the network meta-analysis to some extent.

Conclusions

Point 1: The first conclusion, lines 358 to 360, is not accurate. It is not based on the BBS, but based on the studies using BBS.

Response 1: We have made changes to the original text in response to your suggestions.

Point:Extensive editing of English language and style required

Response:As per your suggestion, we have used the English language touch-up service of MDPI Magazine.

Reviewer 3 Report

The following is simply an observation. Considering that multiple sclerosis (MS) supposedly has a low prevalence in far East areas of the world, it would be interesting to know how rehabilitation procedures are designed in Chinese MS populations. The paper addresses exclusively western medicine approaches but Eastern Asian rehabilitation medical techniques probably deserve to be studied and commented in further projects. Perhaps there were not enough publications on such to be considered in this analysis. 

Author Response

Dear Reviewer,

Thank you for your review, the TCM rehabilitation techniques you mentioned are indeed very interesting, but we did not include them in our study due to the small number of RCTs using TCM rehabilitation techniques to treat multiple sclerosis disease. If there is an opportunity to conduct a study in the future, we will do so in the first instance.

Round 2

Reviewer 1 Report

Thanks for the thorough review, I can only recommend softening the conclusions as much as possible in light of the complex meta-analytic assessment: 

L27 we can suggest

Author Response

Dear Reviewer 1,

Thank you very much for your careful handling of our article and your scientific advice. Based on your suggestions, we have added the appropriate references and made appropriate changes to the conclusion section to make it seem less absolute.

Response to Reviewer 1 Comments

Point 1: Thanks for the thorough review, I can only recommend softening the conclusions as much as possible in light of the complex meta-analytic assessment: 

L27 we can suggest

Response 1:

  • In the conclusions section, we have made appropriate changes. Instead of looking absolutely only at those exercise interventions that achieved first place in the SUCRA rankings, we discuss the results of the NMA more relativistically.
  • We have changed the word "state" to "suggest" as you suggested, which does make the conclusions seem more scientific and less absolute.
  • Finally, we have revisited your suggestions from the first round of reviews, citing the literature you recommended.
